# The Bond Strength and Antibacterial Activity of the Universal Dentin Bonding System: A Systematic Review and Meta-Analysis

**DOI:** 10.3390/microorganisms9061230

**Published:** 2021-06-06

**Authors:** Louis Hardan, Rim Bourgi, Carlos Enrique Cuevas-Suárez, Maciej Zarow, Naji Kharouf, Davide Mancino, Carlos Fernández Villares, Dariusz Skaba, Monika Lukomska-Szymanska

**Affiliations:** 1Department of Restorative Dentistry, School of Dentistry, Saint-Joseph University, Beirut 1107 2180, Lebanon; louis.hardan@usj.edu.lb (L.H.); rim.bourgi@net.usj.edu.lb (R.B.); 2Dental Materials Laboratory, Academic Area of Dentistry, Autonomous University of Hidalgo State, Circuito Ex Hacienda La Concepción S/N, San Agustín Tlaxiaca 42160, Mexico; cecuevas@uaeh.edu.mx; 3Private Practice, “NZOZ SPS Dentist” Dental Clinic and Postgraduate Course Centre—pl. Inwalidow 7/5, 30-033 Cracow, Poland; dentist@dentist.com.pl; 4Department of Biomaterials and Bioengineering, INSERM UMR_S 1121, Biomaterials and Bioengineering, 67000 Strasbourg, France; dentistenajikharouf@gmail.com (N.K.); davidemancino@icloud.com (D.M.); 5Department of Endodontics, Faculty of Dental Medicine, Strasbourg University, 67000 Strasbourg, France; 6Private Practice, Cipriano Sancho, 11 Madrid, 28017 Madrid, Spain; dentistas@azerovillares.com; 7Department of Periodontal Diseases and Oral Mucosa Diseases, Faculty of Medical Sciences in Zabrze, Medical University of Silesia, 40-055 Katowice, Poland; dskaba@sum.edu.pl; 8Department of General Dentistry, Medical University of Lodz, 251 Pomorska St., 92-213 Lodz, Poland

**Keywords:** antibacterial agents, antibacterial properties, bond strength, dentin, universal adhesives

## Abstract

*Streptococcus mutans* (*S. mutans*) is a group of viridans mostly located in oral flora among the wide and biodiverse biofilm. It plays a significant role not only in caries formation but also triggering intracerebral haemorrhage. The durable and stable bond interface, besides bacteria elimination, is one of the crucial factors influencing the resin composite restoration performance. This study aimed to evaluate universal adhesives (UAs) with regard to in vitro bond strength to dentin, and the inhibition of the *S. mutans* growth and compare them with UAs modified with antimicrobial agents through a systematic review and meta-analysis. Two reviewers performed a literature search up to April 2021 in 5 electronic databases: PubMed MedLine, Scielo, ISI Web of Science, Scopus, and EMBASE. Only in vitro studies reporting the effect of modifying UAs with antimicrobial agents on the bond strength to dentin and/or on the inhibition of the *S. mutans* were included. Analyses were carried out using Review Manager Software version 5.3.5 (The Nordic Cochrane Centre, The Cochrane Collaboration, Copenhagen, Denmark). The methodological quality of each in vitro study was evaluated following the parameters of a previous systematic review. A total of 1716 potentially relevant publications were recognized. After reviewing the title and abstract, 16 studies remained in the systematic review. From these, a total of 3 studies were included in the meta-analysis. Since data from the studies included in the antimicrobial outcome included zero values, they could not be meta-analysed. Including 0 values in the analysis will lead to several biases in the analysis, so these data were discarded. The antibacterial effect against *S. mutans* of UAs modified with antimicrobial agents was higher than the non-modified adhesive systems. Within the limitations of the present study, the bond strength of UAs to dentin could be improved by using antimicrobial agents. The UAs modified with antibacterial agents showed a decrease in the viability of *S. mutans* biofilm, among the adhesives tested. However, there are not enough valid data on antibacterial properties of modified UAs; therefore, more well-designed research on these materials is needed.

## 1. Introduction

The clinical success relies on the resistant and durable composite-tooth interface. Dental bonding adhesives (DBAs) create a hybrid layer which fulfils the occurrence of the micromechanical retention of the restoration [1,2]. Residues of bacterial origins left on the cavity surface may damage the adhesive interface. It is important to consider that during carious removal and just before bonding procedures, bacteria can remain within the dentin substrate, smear layer, in dentinal tubules, and at the dentin–enamel junction [3,4]. Hence, it is of critical importance to apply adhesives possessing good antibacterial properties on the cavity surface. Inconsistent findings are present in literature discussing the antibacterial activity of DBAs [5,6,7,8]. Few adhesives exhibit antibacterial properties [6,9]. The application of adhesive systems with no antimicrobial agents can negatively affect clinical outcomes [1]. Various factors affect the antibacterial activity of DBAs, mainly acidity and composition [6]. Adhesion promoting, acidic monomers—containing acrylic, carboxylic, or phosphoric portions in the molecules—are additional factors found to impact the antibacterial activity of the adhesive systems or primers [10].

Nowadays, the latest versions of the so-called universal adhesives (UAs) might be a potential new trend for dentists as they can offer a simplified version of the classical concept of adhesive technology in terms of lessening technique sensitivity and reduced clinical application time [11,12]. This type of adhesives presented various applicability options and it can be used following either a self-etch, or an etch-and-rinse, or a selective etch-enamel mode; therefore, it is also named as multi-mode adhesives [13]. Clinically, however, information regarding their bond performance to dentin is scarce [14]. When dentin bond is concerned, the bonding efficiency of UAs is still debatable; hence, the stability and durability of the dentin–adhesive interface is restricted [15]. Unfortunately, UAs are not able to fully prevent the occurrence of micro-gaps at the adhesive interface [16]. Furthermore, they are also associated with the increased nanoleakage resulting from a mixture of hydrophobic and hydrophilic species in a single bottle DBA [17,18].

The shrinkage of adhesive and resin composite accompanying the polymerization process leads to the formation of microcracks between dentin and adhesive, weakening the bond between them [19,20]. This opens the pathway for the cariogenic bacteria present in saliva such as *S. mutans*. Consequently, microorganisms are able to adhere to the dentinal surface and to reproduce in these microcracks, producing an acidic media which can be mainly responsible for recurrent caries formation, and hypersensitivity [21,22].

It is well known the fact that *S. mutans* is one of the most abundant bacteria found in the oral cavity [23]. This bacterium is responsible for the formation of dental caries and many systemic diseases including intracerebral haemorrhage [24]. In the oral flora, biofilm is formed due to interactions of high complexity between microorganisms, sugar-rich diet, and the host, producing acids that demineralize the dental substrate [25]. Biofilms are defined as microbial communities that are immersed in a three-dimensional extracellular matrix (EM) that are capable of attaching to surfaces. *S. mutans* is still considered the main producer of EM in dental biofilms [26]. It is a highly acidogenic and aciduric microorganism that encodes glucosyltransferases (Gtfs), which leads to the production of extracellular polysaccharides in the presence of sucrose. Extracellular polysaccharides are the primary constituent of the EM and have the ability to provide a framework which supports the biofilm development while encouraging microbial adhesion to surfaces [27].

The introduction of antibacterial agents into dental adhesive may solve the problem of cariogenic bacterial growth in adhesive layer [28,29]. Therefore, the incorporation of the antibacterial quaternary ammonium methacrylate MDPB (12 methacryloyloxydodecylpyridinium bromide) [30], nisin peptide [31], dimethylaminododecyl methacrylate (DMADDM) [32], glutaraldehyde [33], chlorhexidine [34], and silver nanoparticles in dental adhesives [35] might provide a novel perspective on this recurring clinical challenge [36].

These compounds seem to be successful in improving the properties of DBAs by enhancing their long-term performance and protecting the tooth–adhesive interface from microleakage [37]. Due to the limited number of UAs with antibacterial properties on the market, there is a need to investigate compounds that could potentially be added into this novel dental adhesive [38].

A lot of effort had been devoted to assessing the antibacterial activity of UAs. However, there are insufficient numbers of articles on UAs modified with anti-bacterial agents [16,36]. Interestingly, there have been numerous new molecules and methods introduced by several authors in an attempt to attain stable and optimal adhesion of UA agents to dentin substrate [28,29,30,31,32,33,34,35,36,39,40]. Nevertheless, the complex analysis of modified UAs is missing, which could indicate a gold standard for adhesion to dental substrate.

Therefore, the aim of this systematic review and meta-analysis was to evaluate universal adhesives with regard to in vitro bond strength to dentin, and the inhibition of the *S. mutans* growth, and compare them with universal adhesives modified with antimicrobial agents. The null hypothesis of the study was that both bond strength and bacteria inhibition of these adhesives were comparable.

## 2. Materials and Methods

This systematic review and the meta-analysis were executed in accordance with the PRISMA 2020 guidelines [41]. The following PICOS framework was used: population, human dentin; intervention, application of universal adhesives modified by antimicrobial agents; control, application of universal adhesives according to the instructions of the manufacturer; outcomes, growth of *S. mutans* and bond strength; and study design, in vitro studies. The research question was: “Does the modification of universal adhesive systems by antimicrobial agents inhibit the growth of *Streptococcus mutans*, and improve the bond strength to dentin?”

### 2.1. Literature Search

The literature search was independently performed by two reviewers (C.E.C.-S. and R.B.) up to 18 April 2021.The following five electronic databases were screened: PubMed MedLine, Scielo, ISI Web of Science, Scopus, and EMBASE, to identify articles that could be included. The search strategy and keywords used in PubMed are listed in Table 1. The full search strategy for Scielo, ISI Web of Science, Scopus, and EMBASE databases was presented as Appendix A. Respectively, the reviewers also hand-searched the reference lists of included manuscripts for identification of supplementary papers. Following the initial screening, all articles were imported into Mendeley Desktop 1.17.11 software (Glyph & Cog, LLC, London, UK) to remove duplicates.

### 2.2. Study Selection

Two independent reviewers (L.H. and R.B.) evaluated the titles and abstracts of all the manuscripts. Studies for full-text review were selected according to the eligibility criteria: (1) in vitro studies reporting the effect of modifying UAs with antimicrobial agents on the bond strength to dentin, and/or on the inhibition of the *S. mutans*; (2) assessing the bond strength of UAs to dentin substrate with a resin-based material as an antagonist; (3) including a control group in which UAs systems were used following the instructions of manufacturers; (4) including mean and standard deviation data in MPa on micro-shear, shear, micro-tensile, and tensile bond tests; (5) evaluating the bacterial activity of *S. mutans*; (6) publishing in the English language. Papers that involved substrates further than those established in the inclusion criteria were not considered for this review. Clinical trials, case series, case reports, pilot studies, and reviews were also excluded. Full copies of all of the potentially pertinent manuscripts were analysed. Those that seemed to meet the inclusion criteria or had insufficient data in the title and abstract to make a clear decision were designated for full evaluation. The full-text papers were assessed in duplicate by two independent investigators. Any disagreement or variations in view concerning the eligibility of the included manuscripts was resolved and decided through consensus and discussion by a third reviewer (C.E.C.-S.). Only studies that fulfilled all of the eligibility criteria listed were included for review.

### 2.3. Data Extraction

Data of interest from the manuscripts involved were tabulated using a standardized form in the Microsoft Office Excel 2019 spreadsheets (Microsoft Corporation, Redmond, WA, USA). These data contained demographic data (year of publication), universal adhesive, antimicrobial agents, outcomes, and main results. If any information was partially missing, the corresponding authors of the involved papers were contacted twice via e-mail to retrieve the missing data. If authors did not answer within 3 weeks after the first communication, the missing information was not included. For the articles that displayed the information in graph formatting and for which the original data could not be retrieved from the investigators, mean and standard deviation were obtained by calculation using WebPlotDigitizer 4.0 software (Austin, TX, USA).

### 2.4. Quality Assessment

The methodological quality of each involved in vitro study was evaluated by two reviewers (R.B. and L.H.), agreeing to the parameters of the previous systematic review [39]. The risk of bias in each article was assessed regarding the description of the subsequent parameters: specimen randomization; single-operator protocol implementation; blinding of the testing machine operator; the inclusion of a control group; standardization of the sample preparation; failure mode evaluation; use of all the materials following the instructions of the manufacturers; and description of the sample size calculation. If the parameter tested was described by the author, the paper received a “YES” for that particular parameter. On the other hand, when information was missed, the specific parameter received a “NO.” Risk of bias was evaluated and classified by the sum of the “YES” answers received: 1 to 3 designated a high bias, 4 to 6 medium, and 7 to 8 implied a low risk of bias.

### 2.5. Statistical Analysis

Meta-analyses were performed using a software program (Review Manager v5.4.1; The Cochrane Collaboration, Copenhagen, Denmark). The analyses were performed using the random-effects model, and pooled effect estimates were obtained by comparing the standardized mean difference between bond strength values of unmodified universal adhesives versus the antibacterial containing universal adhesive. In studies where several experimental groups were compared with the same control group, data from the experimental groups (mean, standard deviation, and sample size) were combined [42]. A *p*-value < 0.05 was considered statistically significant. Statistical heterogeneity of the treatment effect among studies was assessed using the Cochran Q test and the inconsistency I^2^ test.

## 3. Results

A total of 1716 publications were identified from all databases. A flowchart that outlines the study selection process agreeing to the PRISMA Statement is displayed in Figure 1. The literature review retrieved 1609 publications for the initial examination after the duplicates were removed. Next, 1593 studies were excluded after reviewing the titles and abstracts, leaving a total of 16 studies [33,37,43,44,45,46,47,48,49,50,51,52,53,54,55,56] to be examined by full-text reading. Of these, 13 studies were not included in the qualitative analysis [33,37,43,44,46,47,48,49,51,52,54,55,56], totalizing 3 articles [45,50,53]; the reasons for exclusion are mentioned in the Appendix A.

The characteristics of the manuscripts included in this systematic review are summarized in Table 2. Several antimicrobial agents including fluorinated graphene, eugenyl methacrylate (EgMA), 0.2% chlorhexidine, benzalkonium chloride (BAC) were used as bonding modifiers, while tt-farnesol, ozone, 2% chlorhexidine, resveratrol/ethanol solution, 6% sodium hypochlorite (NaOCl), 0.01% urushiol, epigallocatechin-3-gallate (EGCG), and the mixture dimethyl sulfoxide (DMSO) with epigallocatechin-3-gallate (EGCG) were used as dentin pre-treatments.

Since data from the studies included in the antimicrobial outcome included zero values, they could not be meta-analysed. Including 0 values into the analysis will lead to several biases in the analysis, so these data were discarded.

The meta-analysis suggested that the bond strength of the UAs modified with antibacterial materials (Table 3) was statistically significantly higher when compared with the unmodified UAs (*p* = 0.04). However, a high heterogeneity was observed (I^2^ = 92%) (Figure 2).

Considering the parameters of methodological quality assessment, most of the papers included were scored with medium risk of bias (Table 4). However, several studies analysed failed to report the sample size calculation, single operator, and operator blinded parameters.

## 4. Discussion

This systematic review and meta-analysis were conducted to evaluate UAs with regards to in vitro bond strength to dentin, and the inhibition of the *S. mutans* growth in comparison to UAs modified with antimicrobial agents. The hypothesis that the incorporation of antibacterial components into UAs will not enhance the antibacterial activity of the materials could not be tested. This could be explained by the fact that all the studies included in this systematic review produced a clear inhibition zone against *S. mutans*. However, the bond strength to dentin was improved when the antimicrobial agents were used. Considering what was stated, the null hypothesis proposed in this study was rejected.

The longevity of a restoration can be predicted, to some extent, by its ability to adhere to dental structures, which, in turn, can be measured by bond strength testing [57]. The existing evidence shows that clinical performance can be predicted by appropriate types of laboratory study results. The correlation between laboratory bond strength and clinical retention rates, especially for class V restorations, may be clearly indicated [58]. Hence, bond strength can be assessed in vitro, and it can be measured statically using a macro- or micro-test set-up, basically depending upon the size of the bond area, including macro- and micro-shear bond strength test, macro- or micro-tensile bond strength test, push-out, and pull-out bond strength tests [57].

The improvement in dentin bond strength related to the addition of antibacterial agents (*p* = 0.04) was found. A promising trend in improvement of DBA was the introduction of antibacterial agents aiming at the prevention of secondary caries. MDPB is a monomer composed of a quaternary ammonium monomer (cationic agent) that exhibits biocidal activity by reacting with the negatively charged cytoplasmic membrane of bacteria irreversibly, damaging them [59]. The monomer copolymerizes with other resin monomers in the polymer matrix, preventing it from leaching out and thus ensuring long-lasting antibacterial action [60]. The active ingredient of MDPB is not actually released, but acts as a contact inhibitor against the bacterium that is in direct contact with the restoration.

The antibacterial activity could be linked to the acidity of the functional monomer 10-MDP or to the elution of particular unpolymerized constituents existent in the adhesive system, which is usually toxic to the bacterial colony [61]. Furthermore, agents with anti-biofilm properties served as an excellent inhibitor of cariogenic virulence, suppressed the growth of *S. mutans*, and compromise the acidogenicity [62].

The incorporation of antibacterial agents into UAs would allow for the control of bacterial contamination such as *S. mutans* [28,29]. Correspondingly, these adhesives might counteract the bacterial colonization in gaps produced by interface degradation and resin shrinkage, preventing secondary caries even in the case of adhesives with optimal dentin bond [63].

Another antibacterial agent used in DBAs is chlorhexidine. Chlorhexidine (CHX) is used as salts (diacetate and digluconate). It has been widely used in oral hygiene products and in preventive dentistry due to its antiseptic action with a broad spectrum including *S. mutans* [64]. Chlorhexidine di(acetate)-containing DBA (Peak Universal Bond) did not show antibacterial action against *E. faecalis*, *L. casei,* and *S. mutans* [33]. This can be due to the fact that chlorhexidine is being trapped in the polymer chain of DBA and cannot be effectively released into the surrounding environment. Moreover, chlorhexidine di(acetate)-containing DBA (Peak Universal Bond) did not have any effect against secondary caries in dentin. However, another study showed antibacterial properties of CHX [54].

Additionally, CHX, as MMP-2, -8 and 9 inhibitor, is able to preserve resin–dentin bonding by eliminating or delaying the collagen fibril degradation within the hybrid layer [33,52]. There is a wide dispute on the influence of CHX on the bond strength to dentin. Some studies are reporting that CHX does not affect the shear bond strength to dentin, while others claim the opposite [44,65,66,67,68,69].

Several antimicrobial agents including fluorinated graphene, eugenyl methacrylate (EgMA), 0.2% chlorhexidine, and benzalkonium chloride (BAC) were used as bonding modifiers in this study. These agents are claimed to disinfect the cavity and inactivate residual microorganisms in the dentin surface below the adhesive interface [70,71]. Furthermore, the antibacterial effect of bonding systems inhibits biofilm development at the marginal area of the adhesive interface and on all the tissues surrounding the restoration [72].

Thus, the need of materials with antibacterial activity is essential [73]. First, the better understanding of the substrate composition that will receive the bonding agents must be adopted. Second, the different chemical composition, the type, and the quality of the functional monomer inside the UA may influence the bonding performance [2].

One should bear in mind that *S. mutans* is one of the fundamental pathogens present in the oral cavity surface [74]. This Gram-positive bacterium plays a crucial role in the development of caries and is observed in almost all carious lesions. It can adhere to hard dental substrates with a strong potential for the accumulation of biofilm due to its high surface energy [75].

Bacterial microleakage has been claimed to be the main cause of pulpal inflammation, necrosis, and the eventual need for endodontic therapy after placement of restoration, and the biological sealing of the prepared dentin is now considered critical for successful restoration [5]. In clinical practice, restorative materials are applied to dentin, which is a biological composite of apatite in a collagen matrix with a fluid-filled tubular structure connecting to the pulp [76]. The etching process removes the smear layer and opens dentinal tubules to facilitate the infiltration of monomers, thus irritating the dentin–pulp complex [77]. Furthermore, the flow of dentinal fluid after the adhesive application suggests that the polymerized adhesive could not completely seal the dentin layer close to the pulp in vital teeth [78]. Unpolymerized monomers may be released out of the polymer matrix and diffuse through the tubules into the pulp, causing damage to pulpal tissues. Therefore, with the release of unpolymerized monomer, the irritation to pulp could persist and cause chronic inflammatory response [79].

To emphasize the results obtained by this study, the observed antibacterial activity of UAs against *S. mutans* might be related to the components that are initially incorporated or modified inside adhesive systems, and by the acidic nature of adhesives. Generally, the low pH exhibited by bonding agents is not appropriate to ensure a reliable bactericidal activity since the acidity of these adhesives can be neutralised by the buffering action of the medium [51]. In the present study, non-modified adhesives showed no inhibition against *S. mutans* in comparison to modified UAs. This can be explained by the fact that the non-modified adhesives possess an antibacterial effect themselves due to their low pH. This action is restricted to 24–48 h, and their acidity is neutralized by their contact with the tooth structure [80].

In addition, it is worth mentioning that the surface roughness of dental materials is an important factor that impacts on the bacterial adhesion and proliferation [81]. In this sense, it has been previously demonstrated that the composition can change the roughness of dental materials, including the adhesive systems [82]. Also, composition of the material could affect the hydrophobicity of the surface, and consequently, the bacterial adhesion. Considering this, the addition of antimicrobial agents to UAs can improve antibacterial properties not only by the nature of the antimicrobial monomer, but also by altering the surface of the material. Consequently, dental material development must focus on improving surface characteristics to lessen the possibility of secondary caries. A good contact between hydrophobic composite resin and bonding agents and hydrophilic tooth substrate is hence of great importance [81]. The ideal hybrid layer is composed of the collagen network infused by polymer providing a durable and stable link. The failure of this interface along with biofilm formations may be the starting point for secondary caries formation. The surface quality of composite resins depends on many aspects, including composite composition, microstructure, degree of conversion, finishing, and polishing procedures. From a clinical point of view, finishing and polishing restoration and its margins (sandpaper discs, rubber wheels, and wheels with diamond paste) are mandatory to minimize biofilm accumulation [83].

In this study, ADT was the most-used test [46,49,51,52]. The method incorporates agar plates inoculated with a standardized inoculum of the microorganism under testing *(S. mutans*). Then, filter paper discs that comprise the tested compound are positioned on the agar surface. UAs are dropped using micropipettes on the paper disks or inside the wells in the agar surface. While Petri dishes are incubated under suitable conditions, the antimicrobial agent diffuses into the agar and inhibits the growth and germination of the test microorganism. The diameter of the inhibition growth zone is measured [84]. This method is used for evaluating the release of antibacterial substances but does not determine whether it has bacteriostatic or bactericidal activity. Moreover, paper disks weaken polymerization, and residual monomers may be released, inhibiting bacterial growth. However, the polymerized UAs exhibit weak or lack of antibacterial action [85].

Clinical studies evaluating this variable are scarce. In this review, the best scientific evidence available regarding the dentinal bonding efficacy of UAs modified with antimicrobial compounds was argued. Caution must be exercised when interpreting these results because high heterogeneity was observed in all the comparisons made. Additionally, little information exists regarding the impact of antibacterial adhesive on the enamel bond strength. Moreover, randomized controlled clinical trials must be conducted for providing better insights into the introduction of antibacterial agents into DBAs and their effect on the clinical success of resin-based restorations. Research should be directed towards testing other antibacterial agents with different concentrations in the adhesives, in a form of a delivery providing controlled release without compromising material properties. The relationship between the chemical composition of UAs and bacterial colonization should be clearly established. In the present study, *S. mutans* was the sole target microorganism tested; therefore, further studies evaluating other cariogenic microorganisms should be performed. It must be emphasized that the main reason for failure of dental restorations is secondary caries. Consequently, it seems that establishing a durable and stable dentin bond interface is crucial for the long-term clinical success of restorative treatment.

## 5. Conclusions

To conclude, within the limitations of the long distance between laboratory studies and clinical randomized evaluations, the current in vitro evidence suggests that the dentin bond performance of universal adhesives could be improved by incorporation of antimicrobial agents. Further research on antimicrobial properties of universal adhesives is needed to provide more comprehensive data.

## Figures and Tables

**Figure 1 microorganisms-09-01230-f001:**
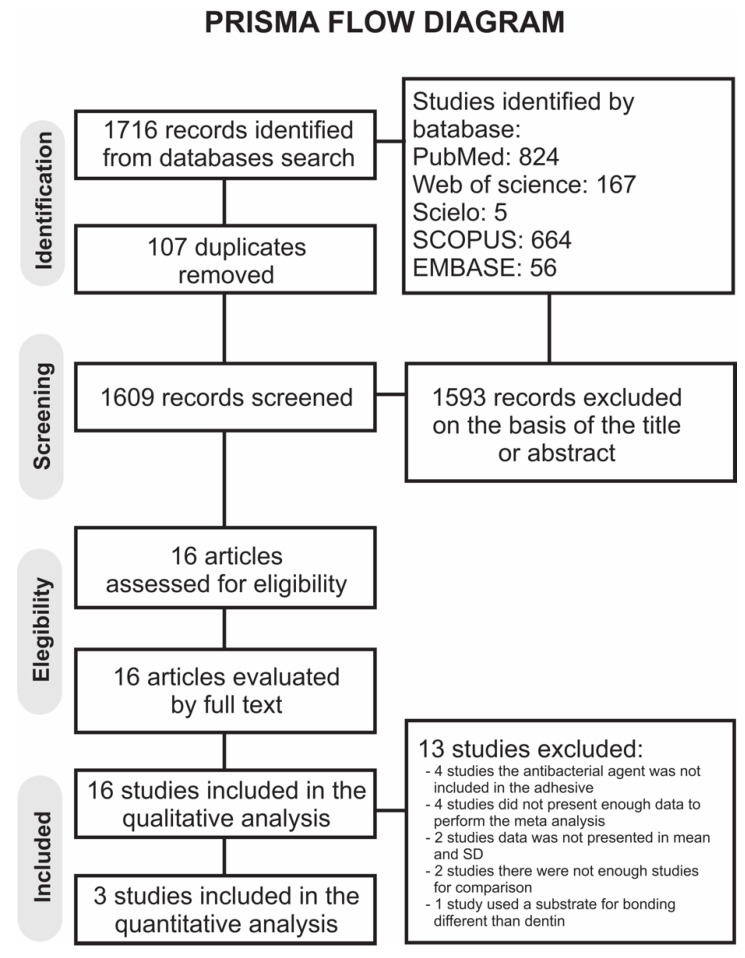
Prisma flow diagram of the study.

**Figure 2 microorganisms-09-01230-f002:**
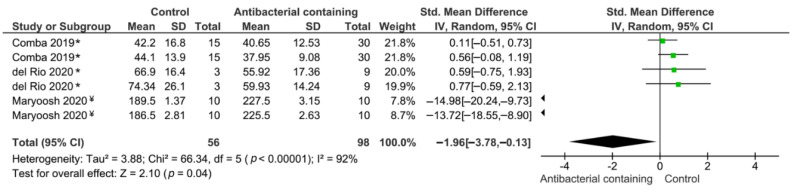
Results of the meta-analysis of the bond strength to dentin of universal adhesives modified with antibacterial agents. * Study used universal adhesives in both self-etch and etch-and-rinse modes; ¥ Study applied universal adhesives in etch-and-rinse mode.

**Table 1 microorganisms-09-01230-t001:** Search strategy used in PubMed.

Search Strategy	
# 1	Microbial viability OR Antibacterial OR Antimicrobial OR peptide antibacterial OR Antibacterial activity OR Anti-Infective Agents OR Anti Infective Agents OR Antiinfective Agents OR Microbicides OR Antimicrobial Agents OR Anti-Microbial Agents OR Anti Microbial Agents OR anti-Bacterial Agents OR Anti Bacterial Agents OR Antibacterial Agents OR Biofilm OR Bacterial
# 2	Universal adhesives OR Universal adhesive OR Universal simplified adhesive systems OR Universal Dental Adhesives OR Multipurpose adhesives OR multi-purpose adhesives OR multimode adhesives OR multi-mode adhesives OR universal bonding agent
# 3	# 1 and # 2

**Table 2 microorganisms-09-01230-t002:** Demographic and study design data of the included studies.

Study	Antimicrobial Agents	Universal Adhesive System	Outcomes	Main Results
Maryoosh 2020	2% fluorinated graphene	Prime & Bond Universal adhesive (Dentsply, Tusla dental specialties, USA) All-Bond Universal adhesive (Bisco Inc., USA)	DC (FTIR) ADT	A significantly greater antibacterial activity was obtained with adhesives containing 2% fluorinated graphene nanoparticles than other groups (*p* < 0.01).
Maryoosh 2020	2% fluorinated graphene nanoparticles	Prime & Bond Universal adhesive (Dentsply, Tusla dental specialties, USA) All-Bond Universal adhesive (Bisco Inc., USA)	Dentin SBS	A higher shear bond strength was observed with adhesives containing 2% FGN in comparison to study groups (*p* < 0.01) after 24 h.
Almaroof 2017	Eugenyl methacrylate (EgMA)	Clearfil Universal Bond (Kuraray, Tokyo, Japan)	DC (FTIR) Glass transition temperature Water sorption and solubility Surface free energy Dentin push-out bond strength SEM CLSM ADT	The total push-out bond strengths of the EgMA-containing adhesives were not significantly different from those of the controls (*p* > 0.05). The modification of the self-etch adhesive system enhanced the bond strength in the middle region of the roots canal. The sizes of the bacterial inhibition zones produced by uncured EgMA modified adhesives were significantly greater (*p* < 0.05) than those of the controls.
Bosso André 2017	0.2% chlorhexidine di(acetate)	Peak Universal Bond (Ultradent Products Inc., South Jordan, UT, USA)	ADT Inhibition of biofilm formation Interface of adhesion	Peak Universal Bond when light cured produced an inhibition halo on S. mutans.
del Rio 2020	tt-farnesol	Adper Scotchbond Universal (3M ESPE, St. Paul, MN, United States)	CFU Biofilm DW Production EIP pH analysis DC Dentin μTBS SEM CLSM	The 3.80% (*v/v*) tt-farnesol-modified adhesive exhibited the lowest CFU count and lowest production of EIP at day 5. Bond strengths decreased with the incorporation of the antibacterial agent into the adhesive system regardless of the concentration of tt-farnesol.
Boutsiouki 2019	0.2% chlorhexidine diacetate 2% chlorhexidine digluconate	Peak Universal Bond (Ultradent Products Inc., South Jordan, UT, USA)	Biological loading in caries model Enamel margins Dentine margins SEM marginal analysis	2% chlorhexidine as dentin pre-treatment, or 0.2% chlorhexidine added in adhesives did not provide any antibacterial effect regarding secondary caries in dentin.
Brambilla 2017	Chlorhexidine diacetate (CDA)	Peak Universal Bond (Ultradent Products Inc., South Jordan, UT, USA)	Tetrazolium salt assay (MTT) CLSM SEM	MTT assay showed that CDA addition decreased, increased or did not change S. mutans biofilm formation. Lowest biofilm formation was obtained with Peak Universal Bond (with and without CDA).
Cangul 2020	Ozone	Peak Universal Bond (Ultradent Products Inc., South Jordan, UT, USA)	Dentin SBS	The application of ozone could be a suitable alternative method to eliminate oral cariogenic bacteria.
Bosso André 2015	0.2% chlorhexidine di(acetate)	Peak Universal Bond (Ultradent Products Inc., South Jordan, UT, USA)	Dentin μTBS DCM DFU	Storage time had no effect on the BS for most of the adhesives. The time required to kill bacteria depended on the type of adhesive and never was less than 10 min.
Peng 2020	Resveratrol/ethanol solution	Scotchbond Universal Adhesive (3 M ESPE, St. Paul, MN, USA)	Dentin μTBS FESEM fractographic analysis Interfacial nanoleakage Surface contact angle test In situ zymography Bacterial culture and biofilm preparation Live/dead bacterial staining MTT assay FESEM examination of biofilm Cytotoxicity evaluation by cell counting Kit-8 (CCK8) assay	The 10 mg/mL resveratrol/ethanol pretreatment group presented significantly higher (*p* < 0.05) μTBS and showed better inhibitory effect of S. mutans activity with acceptable cytotoxicity.
Cha 2016	2% chlorhexidine 6% sodium hypochlorite (NaOCl) 0.01% urushiol	Scotchbond Universal Adhesive (3 M ESPE, St. Paul, MN, USA)	CFU Dentin SBS FE-SEM	All disinfectants tested had strong antibacterial capacity and may better be rinsed away.
Comba 2019	Benzalkonium chloride (BAC)	All-Bond Universal (Bisco Inc.)	Gelatin zymography in situ zymography Dentin μTBS	BAC-containing adhesives reduce endogenous enzymatic activity both immediately and over time, and decrease the bond strength.
Barros Silva 2021	Epigallocatechin3gallate (EGCG)	Universal Single-Bond commercial adhesive (3M ESPE, St. Paul, MN, USA)	CFU Water sorption Solubility	0.5% EGCG was capable of inhibiting biofilm formation; however, it caused significant alteration of the solubility and sorption of the adhesive.
Atalayin 2018	35% Phosphoric acid (Ultra-Etch, Ultradent Products Inc., South Jordan, UT, USA) 37% Phosphoric acid with BAC (Etch-37, Bisco Inc., Schaumburg, IL, USA) 0.2% chlorhexidine diacetate	Peak Universal Bond (Ultradent Products Inc., South Jordan, UT, USA)	ADT	Benzalkonium chloride added into etchant, and chlorhexidine added into adhesive, did not provide additional antibacterial activity against S. mutans.
Zhang 2020	Dimethyl sulfoxide (DMSO) wet-bonding Epigallocatechin-3-gallate (EGCG)	Single-bond Universal (3 M ESPE, St. Paul, MN, USA)	Dentin μTBS Fracture pattern analysis Interfacial nanoleakage evaluation In situ zymography of the hybrid layer Contact angle measurement Antibacterial activity: Live/dead staining of biofilms FESEM observation MTT assay	The synergistic action of DMSO wet-bonding and EGCG can effectively improve dentin–adhesive interface stability. This strategy provides clinicians with promising benefits to achieve desirable dentin bonding performance and to prevent secondary caries, thereby extending the longevity of adhesive restorations.
Kim 2017	2% Chlorhexidine digluconate 6% NaOCl Urushiol	Scotchbond Universal Adhesive (3M-ESPE, St. Paul, MN, USA)	Tooth cavity model Dentin μTBS	The number of S. mutans was significantly reduced in the cavities treated with CHX, NaOCl, and urushiol compared with the control group (*p* < 0.05). However, there was a significant bond strength reduction in the NaOCl group, which showed statistical difference compared to the other groups (*p* < 0.05).

DC: Degree of conversion; ADT: Agar-diffusion test; FE-SEM: Field emission scanning electron microscopic; SEM: Scanning electron microscopy; μTBS: micro-Tensile Bond Strength; μSBS: micro-Shear Bond Strength; CFU: Colony-forming units, DW: dry weight; EIP: extracellular insoluble polysaccharides; CLSM: Confocal laser scanning microscopy CLSM; DCM: direct contact method.

**Table 3 microorganisms-09-01230-t003:** Chemical composition of universal adhesives included in the systematic review.

Material and Manufacturer	Composition *	pH *
Prime & Bond Universal adhesive (Dentsply, Tusla dental specialties, USA)	Bisacrylamide 1, 10-MDP, bisacrylamide 2, DMABN, PENTA, propan-2-ol, water.	2.5
Clearfil Universal Bond (Kuraray, Tokyo, Japan)	Bis-GMA, HEMA, ethanol, 10-MDP, hydrophilic aliphatic dimethacrylate, colloidal silica, dl-camphorquinone, silane coupling agent, accelerators, initiators, water.	2.3
All-Bond Universal adhesive (Bisco Inc, Schaumburg, IL, USA)	Bis-GMA, ethanol, 10-MDP, HEMA.	3.2
Peak Universal Bond (Ultradent Products Inc., South Jordan, UT, USA)	Methacrylic acid, ethyl alcohol, HEMA, chlorhexidine di(acetate)	1.2
Scotchbond Universal Adhesive (3 M ESPE, St. Paul, MN, USA)	Dimethacrylate resins, HEMA, Vitrebond™ Copolymer, Filler, Ethanol, Water, Initiators	2.7

Bis-GMA: Bisphenol A diglycidylmethacrylate; 10-MDP: 10-methacryloxydecyl dihydrogen phosphate; PENTA: dipentaerythritol pentacrylate phosphate; HEMA: 2-Hydroxyethyl methacrylate; DMABN; 4-(dimethylamino)benzonitrile. * According to manufacturers’ Material Safety Data Sheet.

**Table 4 microorganisms-09-01230-t004:** Qualitative synthesis (risk of bias assessment).

Study	Specimen Randomization	Single Operator	Operator Blinded	Control Group	Standardized Specimens	Failure Mode	Manufacturer’s Instructions	Sample Size Calculation	Risk of Bias
Maryoosh	NO	NO	NO	YES	YES	NO	YES	NO	High
Maryoosh	YES	NO	NO	YES	YES	NO	YES	NO	Medium
Almaroof	YES	NO	NO	YES	YES	YES	YES	NO	Medium
Bosso André 2017	YES	NO	NO	YES	YES	NO	YES	NO	Medium
del Rio	NO	NO	NO	YES	YES	YES	YES	NO	Medium
Boutsiouki	YES	NO	NO	YES	YES	NO	YES	NO	Medium
Brambilla	YES	YES	YES	YES	YES	NO	YES	NO	Medium
Cangul	YES	NO	NO	YES	YES	NO	YES	NO	Medium
Bosso André 2015	YES	NO	YES	YES	YES	YES	YES	NO	Medium
Peng	YES	NO	NO	YES	YES	YES	YES	NO	Medium
Cha	YES	NO	NO	YES	YES	NO	YES	NO	Medium
Comba	YES	NO	NO	YES	YES	YES	YES	NO	Medium
Barros Silva	YES	NO	NO	YES	YES	NO	YES	NO	Medium
Atalayin	YES	YES	NO	YES	YES	NO	YES	NO	Medium
Zhang	YES	NO	NO	YES	YES	YES	YES	NO	Medium
Kim	YES	NO	NO	YES	YES	NO	YES	NO	Medium

## Data Availability

The data that support the findings of this study are available from the corresponding author upon reasonable request.

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
