# Peer review of "The Bond Strength and Antibacterial Activity of the Universal Dentin Bonding System: A Systematic Review and Meta-Analysis"

_microorganisms, 2021, doi:10.3390/microorganisms9061230_

Round 1
Reviewer 1 Report
In the article "The Bond Strength and Antibacterial Activity of Universal Dentin Bonding System: A Systematic Review and Meta-Analysis" by Louis Hardan , Rim Bourgi , Carlos Enrique Cuevas-Suárez , Maciej Zarow , Naji Kharouf , Davide Mancino , Carlos Fernández Villares , Dariusz Skaba , Monika Lukomksa-Szymanska, the research theme is interesting from the basic and applied viewpoints.
"The paper represents a comprehensive review for the evaluation of universal adhesives with regard to in vitro bond strength to dentin, and the inhibition of S. mutans growth. When designing the paper, the authors took into account some important factors such as population, human dentin, intervention, application of universal adhesives modified/combined by/with antimicrobial agents, growth of S. mutans in vitro, etc and clearly presented the methodology used for the study selection process. Thus, the information/content presented is meaningful, justifiable, logical, and can be applied to multidisciplinary research domains. From my opinion, this article will be a valuable asset in a rather scarce scientific literature on universal adhesives modified/combined with anti-bacterial agents. "
Author Response
Dear Sir or Madam,
Thank you for your review. We greatly appreciate the effort and time spent to review our article.
In the article "The Bond Strength and Antibacterial Activity of Universal Dentin Bonding System: A Systematic Review and Meta-Analysis" by Louis Hardan , Rim Bourgi , Carlos Enrique Cuevas-Suárez , Maciej Zarow , Naji Kharouf , Davide Mancino , Carlos Fernández Villares , Dariusz Skaba , Monika Lukomksa-Szymanska, the research theme is interesting from the basic and applied viewpoints.
"The paper represents a comprehensive review for the evaluation of universal adhesives with regard to in vitro bond strength to dentin, and the inhibition of S. mutans growth. When designing the paper, the authors took into account some important factors such as population, human dentin, intervention, application of universal adhesives modified/combined by/with antimicrobial agents, growth of S. mutans in vitro, etc and clearly presented the methodology used for the study selection process. Thus, the information/content presented is meaningful, justifiable, logical, and can be applied to multidisciplinary research domains. From my opinion, this article will be a valuable asset in a rather scarce scientific literature on universal adhesives modified/combined with anti-bacterial agents. "
R: We appreciate your kind opinion. Thank you for your comments and time.
Reviewer 2 Report
The authors propose an interesting topic for this review, however it has a few issues which must be addressed before publication.
- The use of English should be checked thought the text by a specialist, especially the use of “the” and plural forms of nouns.
- Abstract:
Line 27: “is one of the crucial factors”
“ This research 46 didn’t receive any external funding. Considering that this systematic review was carried out only 47 in vitro, registration was not executed.” – not necessary in my opinion, can be discarded from the abstract
Line 86 : many systematic systemic diseases including intracerebral haemorrhage [24].
- Discussions are rather long and derivative, consider shortening them where possible.
- The authors should include a paragraph discussing the interactions between the roughness of the materials used, acid and bonding agents used and bacterial adhesion. I suggest analyzing these articles:
Kozmos M, Virant P, Rojko F, Abram A, Rudolf R, Raspor P, Zore A, Bohinc K. Bacterial Adhesion of Streptococcus mutans to Dental Material Surfaces. Molecules. 2021 Jan;26(4):1152.
Taraboanta I, Stoleriu S, Nica I, Georgescu A, Gamen AC, Maftei GA, Andrian S. Roughness variation of a nanohybrid composite resin submitted to acid and abrasive challenges. International Journal of Medical Dentistry. 2020 June 1;24(2):182-87.
- I suggest adding a paragraph discussing the potential damaging effects of these agents on the dental pulp
Author Response
Dear Sir or Madam,
Thank you for your review. We greatly appreciate the effort and time spent to review our article.
The authors propose an interesting topic for this review, however it has a few issues which must be addressed before publication.
- The use of English should be checked thought the text by a specialist, especially the use of “the” and plural forms of nouns.
R: We appreciate your kind opinion. English was revised and corrected by a native speaker.
- Abstract:
Line 27: “is one of the crucial factors”
R: Thank you for your comment. The sentence was corrected.
“ This research 46 didn’t receive any external funding. Considering that this systematic review was carried out only 47 in vitro, registration was not executed.” – not necessary in my opinion, can be discarded from the abstract
R: Thank you for your comment. The sentence was removed.
Line 86 : many systematic systemic diseases including intracerebral haemorrhage [24].
R: Thank you for your comment. The sentence was corrected.
- Discussions are rather long and derivative, consider shortening them where possible.
R: Thank you for the suggestion. Discussion was shortened. Paragraphs describing monomers, antibacterial ingredients and the evaluation of antibacterial properties were deleted.
- The authors should include a paragraph discussing the interactions between the roughness of the materials used, acid and bonding agents used and bacterial adhesion. I suggest analyzing these articles:
Kozmos M, Virant P, Rojko F, Abram A, Rudolf R, Raspor P, Zore A, Bohinc K. Bacterial Adhesion of Streptococcus mutans to Dental Material Surfaces. Molecules. 2021 Jan;26(4):1152.
Taraboanta I, Stoleriu S, Nica I, Georgescu A, Gamen AC, Maftei GA, Andrian S. Roughness variation of a nanohybrid composite resin submitted to acid and abrasive challenges. International Journal of Medical Dentistry. 2020 June 1;24(2):182-87.
R: Thanks for your suggestion. A paragraph was added covering this topic (based on provided references):
‘In addition, worth is mentioning that the surface roughness of dental materials is an important factor that impacts on the bacterial adhesion and proliferation [81]. In this sense, it has been previously demonstrated that the composition can change the roughness of dental materials, including the adhesive systems. [82] Also, composition of the material could affect the hydrophobicity of the surface, and consequently, the bacterial adhesion [83]. Considering this, the addition of antimicrobial agents to UAs can improve antibacterial properties not only by the nature of the antimicrobial monomer, but also by altering the surface of the material.’ – lines 390-397
- I suggest adding a paragraph discussing the potential damaging effects of these agents on the dental pulp
R: Thank you for your comments and time. A paragraph was added covering this topic:
‘Bacterial microleakage has been claimed to be the main cause of pulpal inflammation, necrosis, and the eventual need for endodontic therapy after placement of resto-ration, and the biological sealing of the prepared dentin is now considered critical for successful restoration [5]. In clinical practice, restorative materials are applied to den-tin, which is a biological composite of apatite in a collagen matrix with a fluid-filled tubular structure connecting to the pulp [76]. The etching process removes the smear layer and opens dentinal tubules to facilitate the infiltration of monomers, thus irritating the dentin–pulp complex [77]. Furthermore, the flow of dentinal fluid after the adhesive application suggests that the polymerized adhesive could not completely seal the dentin layer close to the pulp in vital teeth [78]. Unpolymerized monomers may be released out of the polymer matrix and diffuse through the tubules into the pulp, causing damage to pulpal tissues. Therefore, with the release of unpolymerized monomer, the irritation to pulp could persist and cause chronic inflammatory response.[79] ‘ lines 367-379
Round 2
Reviewer 2 Report
The manuscript greatly is improved, however, the discussions section still needs improvements, especially regarding the interaction between adhesive and composite resin interaction, as the composition, chemical structure, and type of surface finishing and polishing method are very important in future bacterial adhesion.
Author Response
Dear Sir or Madam,
Thank you for your review. We greatly appreciate the effort and time spent to review our article.
The manuscript greatly is improved, however, the discussions section still needs improvements, especially regarding the interaction between adhesive and composite resin interaction, as the composition, chemical structure, and type of surface finishing and polishing method are very important in future bacterial adhesion.
R: Thank you for your suggestion. A paragraph was added covering this topic: ‘Consequently, dental material development must focus on improving surface characteristics to lessen the possibility of secondary caries. A good contact between hydrophobic composite resin and bonding agents and hydrophilic tooth substrate is hence of great importance [81]. The ideal hybrid layer is composed of the collagen network infused by polymer providing a durable and stable link. The failure of this interface along with biofilm formations may be the starting point for secondary caries formation. The surface quality of composite resins depends on many aspects including composite composition, microstructure, degree of conversion, finishing, and polishing procedures. From a clinical point of view, finishing and polishing restoration and its margins (sandpaper discs, rubber wheels, and wheels with diamond paste) is mandatory to minimize biofilm accumulation [83]. ‘ lines 332-342